# Dosage suppressors of *gpn2^{ts}* mutants and functional insights into the role of Gpn2 in budding yeast

**Le Wang**[1][☯], **Pan Li**[2][☯], **Pei Zeng**[1][☯], **Debao Xie**[1], **Mengdi Gao**[1], **Lujie Ma**[1], **Aamir Sohail**[1], **Fanli Zeng**[1]*

**1** College of Life Sciences, Hebei Agricultural University, Baoding, Hebei, China, **2** College of Plant Protection, Hebei Agricultural University, Baoding, Hebei, China

☯ These authors contributed equally to this work.
* fanli.zeng@pku.edu.cn

**Data Availability Statement:** All relevant data are within the manuscript and its Supporting Information files.

## Abstract

Gpn2 is a highly conserved protein essential for the assembly of RNA polymerase II (RNA-PII) in eukaryotic cells. Mutations in Gpn2, specifically Phe105Tyr and Leu164Pro, confer temperature sensitivity and significantly impair RNAPII assembly. Despite its crucial role, the complete range of Gpn2 functions remains to be elucidated. To further explore these functions, we conducted large-scale multicopy suppressor screening in budding yeast, aiming to identify genes whose overexpression could mitigate the growth defects of a temperature-sensitive *gpn2* mutant (*gpn2^{ts}*) at restrictive temperatures. We screened over 30,000 colonies harboring plasmids from a multicopy genetic library and identified 31 genes that rescued the growth defects of *gpn2^{ts}* to various extents. Notably, we found that *PAB1*, *CDC5*, and *RGS2* reduced the drug sensitivity of *gpn2^{ts}* mutants. These findings lay a theoretical foundation for future studies on the function of Gpn2 in RNAPII assembly.

## Introduction

Gpn2 is a conserved member of the Gpn-loop GTPase family, which includes essential proteins Gpn1 (Npa3) and Gpn3. This family, originating from a single precursor gene in archaea, exhibits significant sequence divergence yet retains distinct functional roles [1]. In yeast, functional defects in Npa3, Gpn2, or Gpn3, as well as the knockdown of Gpn1 or Gpn3 in human cells, result in the cytoplasmic accumulation of RNA polymerase II (RNAPII) subunits Rpb1 and Rpb3. This suggests a direct involvement of these proteins in RNAPII assembly [2–8]. Particularly, Gpn2 has been identified as critical for the assembly of the Rpb3 subcomplex during RNAPII biogenesis in budding yeast [9]. However, the full spectrum of Gpn2 functions remains unclear.

In previous studies, Gpn2 was identified as a key factor in RNAPII assembly, directly interacting with the Rpb12 and Rpb3 subunits, as well as the Rba50 assembly factor [9]. At the same time, genetic screening further confirmed the association of Gpn2 with Rpb12, Rpb3 and Rba50 [9]. The *gpn2^{ts}* mutant demonstrates temperature sensitivity and is susceptible to the

**Funding:** This work was supported by grants from the National Key Research and Development Program of China (2023YFD1401500 to F.Z.), the National Natural Science Foundation of China (no. 32070574 to F.Z.), the Natural Science Foundation of Hebei Province (C2024204178 to F.Z.), and Operation Expenses for Universities' Basic Scientific Research of Hebei Province (KY2024043 to F.Z.).

**Competing interests:** The authors declare that they have no known competing financial interests or personal relationships that could have appeared to influence the work reported in this paper.

transcription inhibitor mycophenolic acid (MPA) [9]. Furthermore, the Gpn2-Rba50 complex is conserved and essential for cellular functional viability. Deficiencies in Gpn2 and Rba50 activity disrupt the assembly of the Rpb3 subcomplex, adversely affecting subsequent stages in the assembly pathway. Notably, our findings reveal that Gpn2 promotes and regulates the integration of Rpb2 with Rba50. The Gpn2-Rba50 complex also plays a similar role in the assembly of RNAP III [10].

The power of unraveling novel functional relationships through genetic interactions, such as gene-gene or epistatic interactions, is well-documented. Suppression is a quintessential genetic interaction that facilitates the discovery of novel interactions and the identification of new genes [11, 12]. Multicopy genetic screening stands as one of the most effective high-throughput methods for identifying dosage suppressors [13, 14]. In yeast, multicopy libraries are commonly utilized to isolate genes that confer suppression phenotypes through elevated expression levels [15]. This screening method can simultaneously identify multiple genes related to a query gene, thereby unveiling new gene functions and providing insights into the roles of previously uncharacterized genes. Such multicopy genetic screens have proven successful in budding yeast, where it was determined that *RPB12* and *RBA50* act as suppressors for *gpn2^{ts}* mutants. This genetic evidence demonstrates that Gpn2, in conjunction with Rpb12 and Rba50, directly influences assembly of the Rpb3 subcomplex [9]. Similarly, our studies have shown that *RBA50* and *GPN3* serve as multicopy genetic suppressors for *npa3^{ts}* mutants [9].

Based on previous research that identified suppressors such as Rpb12, Rba50, and other GPNs [9], we sought to further elucidate the essential functions of Gpn2 by conducting a systematic large-scale genetic screen for additional suppressors of *gpn2^{ts}* mutants [16]. In this study, we report the identification of 31 suppressor genes that enhance the growth of *gpn2^{ts}* and reveal multiple roles for Gpn2 in budding yeast. Based on the mechanism of genetic suppression, we propose potential functions for previously unidentified suppressors, such as Pab1, Cdc5, and Rgs2, which may act as chaperonins in RNAPII assembly. Additionally, we conducted drug sensitivity assays, which demonstrated that *PAB1*, *CDC5*, and *RGS2* can mitigate the effects of DNA-damaging drugs and transcriptional inhibitors on *gpn2^{ts}* at 32˚C. Furthermore, at 34˚C, *PAB1*, *CDC5*, and *RGS2* each can compensate for *gpn1^{ts}*, with *CDC5* reducing the sensitivity of *gpn1^{ts}* to 4-nitroquinoline oxide (4-NQO) tablets. Through fluorescence localization of Rpb1, we observed that high expression of *CDC5* can compensate for the nuclear dispersion observed in the *gpn2^{ts}* mutant. These findings provide a theoretical foundation for in-depth exploration of the regulatory mechanisms by which Gpn2 is involved in RNAPII assembly.

## Materials and methods

### Yeast strains and media

The *gpn2^{ts}* strains (W303-1a, *gpn2-del::gpn2-F105Y L164P-13myc:URA3*) were utilized for the multicopy genetic suppressor screen assessing temperature sensitivity at 34˚C [9]. Derivative strains were used to evaluate suppression levels and to test sensitivity to DNA-damaging agents. All manipulations were performed using the *Saccharomyces cerevisiae* W303-1a strain (MATa; *ade2-1*; *ura3-1*; *his3-11, 15*; *trp1-1*; *leu2-3, 112*; *can1-100*). The *RPB1* with a C-terminal *GFP* tag was generated at their genomic loci by integration of the PCR-generated cassettes amplified from genomic DNA of the *Yeast GFP Clone Collection*, using oligonucleotides complementary to the sequences flanking the endogenous loci [9].

Yeast-peptone-dextrose (YPD) is a rich medium that contains 1% yeast extract, 2% peptone, and 2% glucose. Synthetic dropout (SD) is a synthetic medium supplemented with

amino acids. Yeast cells were cultured at specified temperatures in YPD, except for those carrying pRS425 (2 μm)-based multicopy constructs, which were grown in leucine-free SD medium (SD-Leu⁻, 0.67% yeast nitrogen base [YNB], 2% glucose, 0.02% tryptophan, 0.02% histidine, 0.02% adenine, and 0.02% uracil). Mycophenolic acid, hydroxyurea, methyl methane sulphonate, and 4-nitroquinoline oxide (Sigma) were added at the indicated concentrations. For ten-fold serial dilution assays, cultures grown exponentially at 25˚C were spotted onto plates in ten-fold serial dilutions and incubated at specified temperatures for three to four days before imaging [10].

## Isolation of multicopy suppressors of *gpn2^ts^*

Approximately $1\times10^8$ log-phase *gpn2^ts^* cells were transformed with 5 μg of plasmid DNA from a 2 μm-based multicopy yeast genomic DNA library. This library, partially digested with Sau3AI, was constructed in the pRS425 vector (2 μm ARS *LEU2*) with an average insert size of approximately 10 kb [16]. After transformation, cells were plated on SD-Leu⁻ plates and incubated at 25˚C for one day, followed by 34˚C for five days. The transformation efficiency was approximately $1\times10^4$ transformants per μg of DNA. Plasmids from the colonies that survived at 34˚C were isolated and transformed into the *E. coli* DH5α strain for amplification. To verify that the suppression function was attributable to the plasmid and not a chromosomal mutation in the original yeast transformant, plasmids were retransformed into the *gpn2^ts^* strain. The inserts were sequenced using the T7 and T3 primers. Subsequently, individual genes were subcloned into pRS425 and retested to confirm their role in the suppression of *gpn2^ts^*.

## Fluorescence microscopy

For the fluorescence localization of Rpb1-GFP, cells harboring pRS425-based constructs were initially cultured in the SD-Leu⁻ medium at 25˚C. These cultures were then shifted to 34˚C for a period of 3 hours. Subsequently, the selected culture was harvested and fixed using 70% ethanol. After fixation, the cells were washed and imaged using a Nikon fluorescent inverted microscope [9].

## Results

### Genetic screening for multicopy suppressors of the *gpn2^ts^* mutant

The rapid degradation of Gpn2 proved non-viable, underscoring its role as an essential protein [4]. Gpn2 contains a GTP-binding domain ranging from amino acids 7 to 261 and is highly conserved across species from archaea to yeast and humans. This evolutionary conservation suggests that Gpn2 may be pivotal in central biological processes (Fig 1A).

Through random mutagenesis of *GPN2,* we isolated a mutant, *gpn2^ts^*, featuring a double mutation (Phe105Tyr and Leu164Pro). Phe105 is situated near the G3 motif, a crucial GTPase-interacting domain essential for GTP binding and hydrolysis, whereas Leu164 resides within the conserved residue loop [9] (Fig 1B). This functionally compromised Gpn2 variant shows temperature sensitivity (Fig 1C) and increased susceptibility to genotoxic stresses such as hydroxyurea (HU), methyl methane sulphonate (MMS), and 4-nitroquinoline oxide (4-NQO) (Fig 1D). *GPN2* was initially characterized as a chromosomal instability (CIN) mutant with defects in sister chromatid cohesion [5, 17]. These findings reinforce the hypothesis that *GPN2* is indispensable for maintaining genomic integrity.

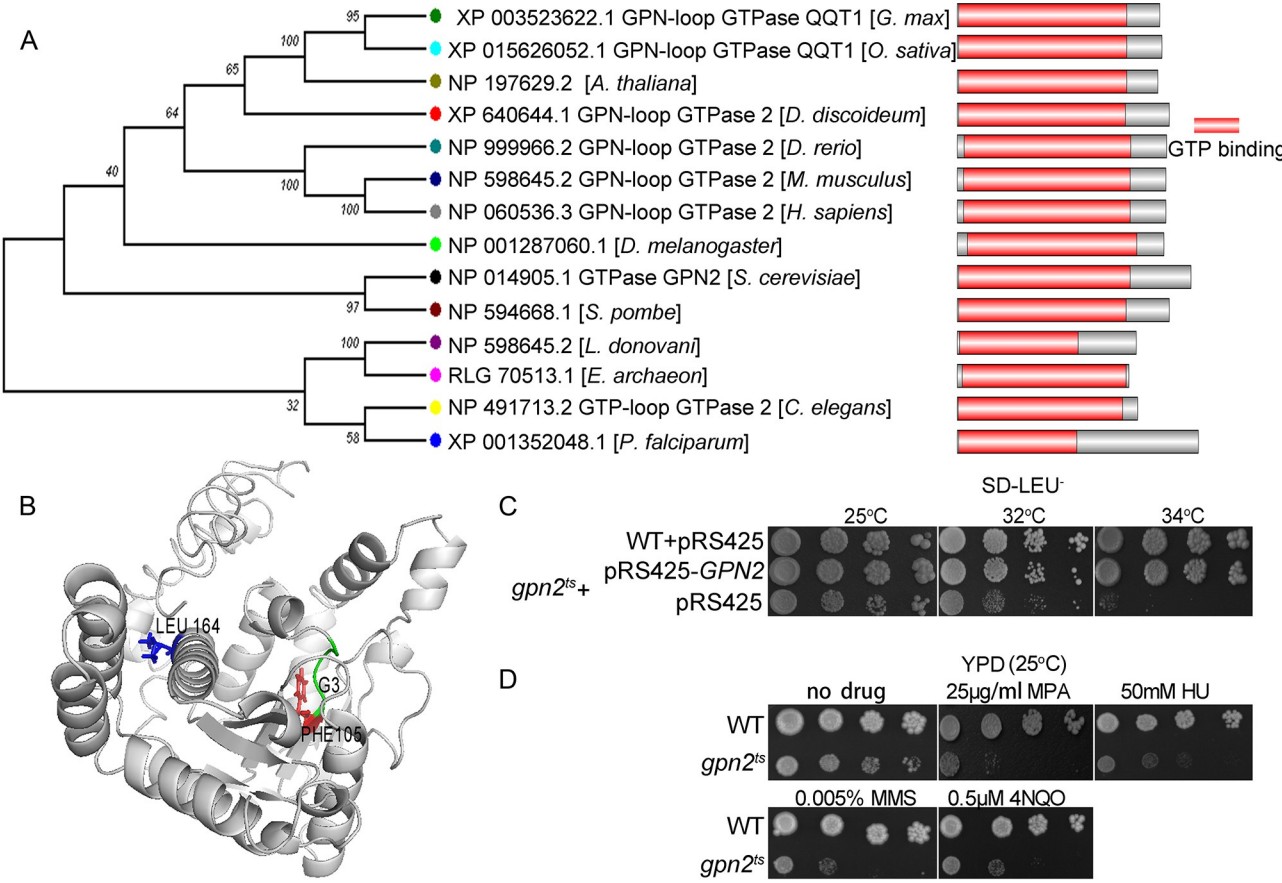

**Fig 1. *GPN2* is an evolutionarily conserved protein.** A: The phylogenetic tree was generated based on the multiple sequence alignment of full-length *GPN2* amino acid sequences using the neighbor-Joining method in MEGA6 with 1000 bootstraps. Pink bars indicate domains predicted using the Pfam website (https://pfam.xfam.org). B: Molecular model of the yeast Gpn2 GTPase domain protein dimer. The *GPN2* model was simulated by Zhanglab (https://zhanglab.ccmb.med.umich.edu). Conserved G3 domains (green) and the Phe105 (red) and Leu164 (blue) mutation sites of *gpn2^ts* are shown. C: The *gpn2^ts* mutant was transformed with multicopy plasmids pRS425 and pRS425-*GPN2*. The W303-1a (WT) strain was transformed with pRS425. These strains were spotted in tenfold serial dilutions on SD-Leu⁻ plates and incubated at the indicated temperatures for four days. D: The indicated strains were spotted in ten-fold serial dilutions on YPD plates containing mycophenolic acid (MPA), hydroxyurea (HU), methyl methane sulfonate (MMS), or 4-nitroquinoline oxide (4-NQO), and incubated for four days at 25˚C.

To delineate the specific biological functions of Gpn2, we performed a large-scale search for multicopy suppressor genes that could mitigate the temperature sensitivity of *GPN2* mutants at restrictive temperatures (refer to the Materials and Methods section) (Fig 2).

We screened over 30,000 colonies, isolating 100 potential suppressors. Plasmids extracted from these colonies were retransformed into DH5α, confirming 60 candidates. These plasmids were sequenced to identify the genomic regions responsible for suppression. Many contained overlapping genomic fragments, collectively representing 31 unique genomic regions. Further subcloning experiments pinpointed specific genes linked to the suppressive phenotype. We identified 31 such genes that varied in their ability to correct the growth defects of *gpn2^ts* and their sensitivity to the transcription inhibitor MPA (Fig 3). These genes span various functions, pathways, and cellular locations, including DNA repair and RNA polymerase biogenesis (Table 1).

## Multicopy genetic suppressors of *gpn2^ts*

The suppressor genes identified can be categorized into different functional groups based on their known functions:

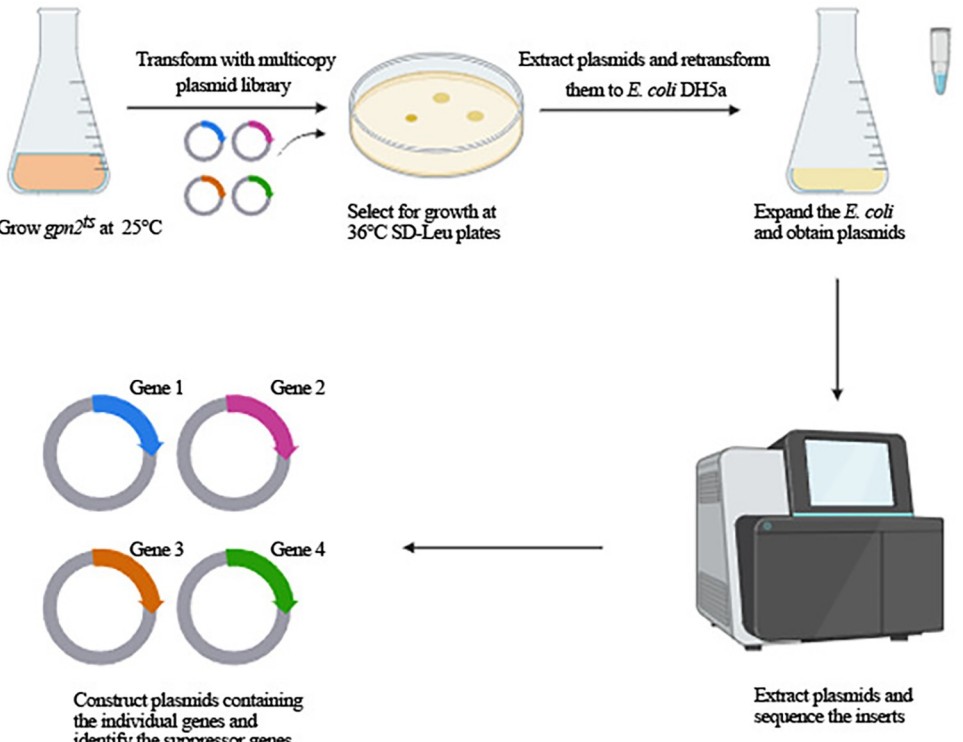

**Fig 2. Genetic screening for multicopy suppressors of the *gpn2^ts* growth defect.** The strategy for isolating multicopy suppressors is described in the 'Materials and Methods' section.

**Genes encoding molecular chaperones.** *LHS1*. Lhs1 acts as a molecular partner to hsp70 within the endoplasmic reticulum lumen, playing a crucial role in peptide translocation and folding. It is regulated by the unfolded protein response pathway [18, 19]. Lhs1 has a homolog, *GRP170*, and recent research indicates that Lhs1 selectively recognizes misassembled forms of the episodic sodium channel (α ENaC) via its transmembrane domain [20, 21].

*NAS2*. Nas2, a member of the *SLC13* gene family, is a Na$^+$-coupled transporter originally isolated from high endothelial venule endothelial cells. It is known for transporting sulfates in conjunction with Na$^+$ and also mediates the transport of oxygen anions from micronutrients such as selenium and chromium [22]. Additionally, Nas2 is an evolutionarily conserved partner in the assembly of the 19S proteasome regulatory granule (RP) base subcomplex [23]. Recent studies have shown that Nas2 plays a pivotal role in the assembly of six different ATPase subunits (Rpt1 to Rpt6) in the proteasome, facilitating one of the final steps in their assembly [24].

*HSP150*. Hsp150, an O-mannosylated heat shock protein, is part of the Pir protein family. It plays a crucial role in cell wall stability and is induced by heat shock, oxidative stress, and nitrogen limitation [25–27]. These chaperones may facilitate the proper folding of Gpn2 or interact with RNA polymerase subunits as assembly chaperones. They could represent novel RNAPII assembly factors, similar to Hsc82/Hsp82 and the R2TP/prefoldin-like complex involved in RNA polymerase assembly [28].

**Genes affecting nuclear transport/splicing.** *PAB1*. Pab1 is the primary poly(A)-binding protein in yeast, playing a pivotal role in mRNA stability, nuclear export, and translation initiation [29, 30]. Its deletion affects various processes, including translation initiation, translation termination, and mRNA decay [31]. Furthermore, Pab1 has been recognized as a key marker

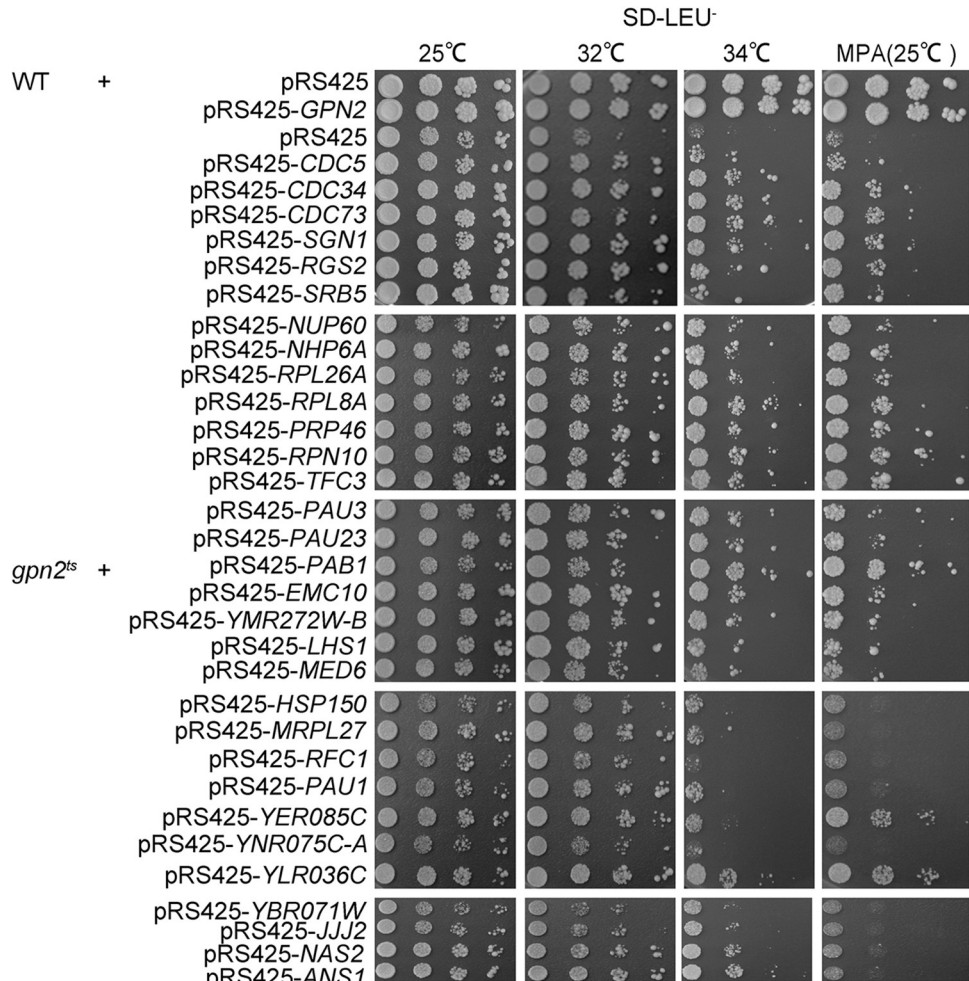

**Fig 3. Multicopy suppressors rescue the *gpn2^ts* growth defect to various extents.** Tenfold serial dilutions of exponentially growing WT cells (W303-1a) carrying pRS425 mock plasmid and *gpn2^ts* mutants carrying the indicated constructs were spotted onto SD-Leu⁻ plates with or without MPA and incubated at the indicated temperatures for four days.

for stress granules, which form through phase separation in response to physiological stress, resulting in the creation of hydrogels outside the cell [32]. We hypothesize that the formation of stress granules serves as a self-protective mechanism for cells.

*NUP60*. Nup60 is an FG-nucleoporin component of the central core of the nuclear pore complex, playing a crucial role in nucleocytoplasmic transport and maintaining the permeability barrier of the nuclear pore complex (NPC) [33, 34]. This study suggests that Gpn2, known as an RNAPII assembly factor, may also influence the nuclear transport of RNAPII, as evidenced by Gpn1 and Gpn3 acting as multicopy suppressors for the *gpn2^ts* mutant [3, 35–38].

*PRP46*. Prp46, a member of the NineTeen Complex (NTC), is vital for pre-mRNA splicing [39, 40]. It may compensate for defective Gpn2 by altering splice-site recognition or promoting alternative splicing [41].

**Transcription-/translation-related genes.** *TFC3*. Tfc3 is a subunit of the RNA polymerase III transcription initiation factor complex (*TFIIIC*) in *Saccharomyces cerevisiae*. Composed of six distinct subunits, *TFIIIC* binds to RNA polymerase III promoters, initiating the assembly of the transcription complex. The largest yeast subunit, tau138, encoded by *TFC3*, targets the

**Table 1. Suppressor genes identified in the screen.**

| Gene Name | Short Description | Degree of Suppression |
|---|---|---|
| CDC34 | Ubiquitin-conjugating enzyme (E2) and catalytic subunit of SCF ubiquitin-protein ligase complex | Strong |
| CDC73 | Component of the Paf1p complex and telomere maintenance | Strong |
| SGN1 | Cytoplasmic RNA-binding protein | Strong |
| RPL8A | Ribosomal 60S subunit protein L8A | Strong |
| PRP46 | Member of the nine teen complex (NTC) | Strong |
| RPN10 | Proteasome polyubiquitin receptor and non-ATPase subunit of the 19S regulatory particle (RP) of the 26S proteasome | Strong |
| TFC3 | Subunit of RNA polymerase III transcription initiation factor complex | Strong |
| PAB1 | Poly(A) binding protein, part of the 39-end RNA-processing complex | Strong |
| YLR036C | Putative protein predicted to have transmembrane domains | Strong |
| ANS1 | Putative GPI protein; *SWAT*-GFP and mCherry fusion proteins localize to the vacuole | Strong |
| CDC5 | Polo-like kinase | Medium |
| RGS2 | Negative regulator of glucose-induced cAMP signaling | Medium |
| NUP60 | FG-nucleoporin component of central core of the nuclear pore complex | Medium |
| NHP6A | High-mobility group non-histone chromatin protein | Medium |
| RPL26A | Ribosomal 60S subunit protein L26A | Medium |
| PAU3 | Member of the seripauperin multigene family | Medium |
| PAU23 | Cell wall mannoprotein; has similarity to Tir1p, Tir2p, Tir3p, and Tir4p | Medium |
| EMC10 | Putative protein of unknown function | Medium |
| YMR272W-B | Protein of unknown function | Medium |
| JJJ2 | Protein of unknown function | Medium |
| NAS2 | Evolutionarily conserved 19S regulatory particle assembly-chaperone | Medium |
| SRB5 | Subunit of the RNA polymerase II mediator complex | Weak |
| LHS1 | Molecular chaperone of the endoplasmic reticulum lumen and nucleotide exchange factor for the ER lumenal Hsp70 chaperone Kar2p | Weak |
| MED6 | Subunit of the RNA polymerase II mediator complex | Weak |
| HSP150 | O-mannosylated heat shock protein | Candidate suppressor |
| MRPL27 | Mitochondrial ribosomal protein of the large subunit | Candidate suppressor |
| RFC1 | Subunit of heteropentameric replication factor C (RF-C) | Candidate suppressor |
| PAU1 | Member of the seripauperin multigene family | Candidate suppressor |
| YER085C | Putative protein of unknown function | Candidate suppressor |
| YNR075C-A | Protein of unknown function | Candidate suppressor |
| YBR071W | Protein of unknown function found in the cytoplasm and bud neck | Candidate suppressor |

B-box promoter element. Mutations in *TFC3* disrupt tRNA and 5S RNA synthesis, underscoring its critical role in transcription [42–44].

*SRB5 and MED6*. Srb5 and Med6 are subunits of the RNAPII mediator complex, crucial for associating with core polymerase subunits to form the RNAPII holoenzyme, essential for transcriptional regulation [45, 46]. Additionally, the abundance of Med6 protein increases in response to DNA replication stress, highlighting its role in cellular stress responses [47, 48].

*SGN1*. *SGN1* encodes a cytoplasmic RNA-binding protein featuring an RNA recognition motif (RRM). Overexpression of *SGN1* has been shown to suppress the temperature-sensitive growth phenotype in specific alleles of *TIF4631*, *TIF4632*, and *PAB1*, all of which are involved in translational initiation. These findings imply that *SGN1* may play a role in mRNA translation [49].

The suppressive effect of these genes likely influences transcription and translation processes. Overexpression may stabilize mRNA or protein levels of *gpn2^ts^*, thereby preserving

their translation and the accumulation of residual active peptides [49]. These observations also raise the possibility that Gpn2 may directly or indirectly impact transcription or mRNA translation.

**Ribosome-related genes.** *RPL26A*. Rpl26a, part of the ribosomal 60S subunit, binds to 5.8S rRNAs. While the deletion of *RPL26A* has minimal effects, overproduction of the yeast ribosomal protein Rpl26 fails to integrate into ribosomes, indicating that protein L26 is not essential for ribosome assembly and function [50, 51]. Additionally, *RPL26A* and *RPL14A* act as weak suppressors of a conditional mutation in the basal transcription factor *TFIIIC*, suggesting a supportive role in transcription under constrained conditions [52].

*MRPL27*. Mrpl27 is a component of the mitochondrial ribosomal large subunit. Mutants with disrupted *MRPL27* genes are unable to grow on non-fermentable carbon sources, demonstrating that *MRPL27* is crucial for mitochondrial functionality in yeast [53].

**Protein regulation-related genes.** *CDC34*. *CDC34* encodes a ubiquitin-conjugating enzyme (E2), serving as the catalytic subunit of the SCF ubiquitin-protein ligase complex. This complex is responsible for targeting proteins for degradation, including those implicated in cell cycle progression. Elevated levels of Cdc34 under DNA replication stress conditions suggest its crucial role in managing genotoxic stress [54, 55].

*RPN10*. Rpn10, a proteasome polyubiquitin receptor, features a ubiquitin-interacting motif that selectively binds to polyubiquitin chains, playing a vital role in intracellular proteolysis [56]. As part of the proteasome regulatory particle, Rpn10 primarily coordinates with the ubiquitin-type *DSK2* receptor to form the Dsk2-proteasome, facilitating its function [57].

*RGS2*. Rgs2 serves as a negative regulator of glucose-induced cAMP signaling, in addition to its role as a general regulator of G protein signaling (RGS). It directly activates the GTPase activity of the heterotrimeric G protein alpha subunit, Gpa2p [58, 59]. Rgs2 may also regulate Gpn2, a GTPase from the GPN family, acting as a GTPase activating protein (GAP). As a multifunctional *RGS* protein, *RGS2* regulates various signaling pathways linked to G proteins. Research interest in *RGS2* extends beyond cancer to include implications in cardiovascular and cerebrovascular diseases.

**Genes affecting genome stability.** *CDC73*. Cdc73 is a key component of the Paf1p complex and interacts with RNA polymerases I and II to modulate their activities. This protein is involved in transcription elongation and regulates ATR activity. Overexpression of *CDC73* mRNA promotes carcinogenesis [60–63]. Additionally, studies indicate that Cdc73 plays a crucial role in maintaining genomic stability by mediating telomere homeostasis [64].

*CDC5*. Cdc5, a Polo-like kinase essential for the mitotic cell cycle, regulates the nuclear shape and the expansion of the nuclear envelope during mitosis. It plays a crucial role in the cellular response to DNA damage by attenuating the DNA damage checkpoint by reducing Rad53 hyperphosphorylation, allowing cells to adapt to DNA damage [65–68]. *CDC5* also interacts with promoters of genes encoding microRNAs (MIRs) and DNA-dependent RNA-PII, influencing MIR promoter activity and RNAPII occupancy. This interaction suggests that high expression of *CDC5* may stabilize RNAPII, thereby mitigating the temperature sensitivity of *gpn2$^{ts}$* [69].

*RFC1*. Rfc1 is a subunit of the heteropentameric replication factor C (RFC), a DNA-binding protein and ATPase essential in DNA repair processes. RFC binds to cell cycle checkpoint proteins, initiating signal transduction downstream of DNA damage checkpoints, and is involved in mismatch repair and the excision repair of damaged DNA [8, 70]. Although *RFC1* has minimal impact on *gpn2$^{ts}$* mutants under restricted temperatures, high expression of *RFC1* at 32˚C can alleviate the growth defects of these mutants, suggesting it may indirectly reduce *gpn2$^{ts}$* sensitivity.

*NHP6A*. Nhp6A is a high-mobility group (HMG) protein that binds to DNA. It facilitates the maintenance of genome stability and plays important roles in transcription and DNA

replication [71, 72]. In yeast, the *NHP6A/B* proteins are crucial for initiating RNA polymerase II transcription. Complementation of these proteins may not only enhance RNA polymerase II transcription but also partially compensate for the deficiencies observed in *gpn2*[ts] mutants [73].

In budding yeast, mutations in *GPN2* have been shown to increase the frequency of chromosome fragment loss, potentially influencing the establishment of sister chromatid cohesion [17]. The defective Gpn2 protein disrupts the assembly of RNAPII, which may directly bind to the single-stranded DNA at the termini of DNA breaks, subsequently transcribing RNA and forming an R-loop [5, 74, 75]. This study demonstrates that overexpression of cell cycle regulatory genes such as *CDC34*, along with DNA repair-related genes, can mitigate the growth defects associated with *gpn2*[ts] mutants. The cell cycle significantly influences the choice of pathway for DNA double-strand break repair [76]. These findings suggest that *GPN2* plays a role in DNA repair processes and impacts genome stability. Given that genomic instability is a hallmark of cancer, *GPN2* may also have potential implications in cancer development.

*Additional genes*. This category encompasses three members of the seripauperin multigene family (*PAU1*, *PAU3*, and *PAU23*), primarily encoded in subtelomeric regions and negatively regulated by oxygen levels (Fig 4D) [77]. It is hypothesized that mutations in *GPN2* may influence intracellular oxygen content. Additionally, this category includes genes of unknown function, such as *EMC10*, *YMR272W-B*, *YER085C*, *ANS1*, *YNR075C-A*, *JJJ2*, *YBR071W*, and *YLR036C*. Our study aims to provide valuable insights into their roles, potentially advancing functional research.

## Possible suppression mechanisms

The potential suppression mechanisms of dosage suppressors of *gpn2*[ts] primarily include: 1) regulating or stabilizing the mutant protein; 2) engaging downstream factors that circumvent the need for the mutant protein. For instance, the *RGS2* gene, which encodes a known GTPase-activating protein (GAP), may regulate or stabilize Gpn2 activity, aligning with the first mechanism. The chaperone proteins Hsp150, Nas2, and Lhs1 are crucial for the proper folding of newly synthesized polypeptides. Their overexpression may enhance the stability of either the Gpn2 mutant protein or essential subunits of RNAPII, fitting into the first suppression category. Additionally, transcription- and translation-related genes, including *TFC3* and *SGN1*, suggest that increasing the levels of mutant *GPN2* mRNA or protein can suppress the observed temperature-sensitive (TS) phenotype, also supporting the first explanation. Lastly, interaction network analysis places the Gpn2-centered pathway in parallel with the Cdc34 pathways, aligning with the second suppression mechanism (Fig 4A).

## *CDC5* is a suppressor of *gpn1*[ts] and *gpn2*[ts]

Multicopy genetic screening, a high-throughput method, is employed to identify dosage suppressors [13], enabling the simultaneous discovery of multiple genes related to a query gene, aiding in the discovery of new gene functions and providing insights into the functions of unknown genes. In our study, we selected the nuclear transporter Pab1 [29], the genome stability-related protein Cdc5 [67], and the G protein signaling regulator Rgs2 to investigate their roles in the assembly of RNAPII [59]. Our findings indicate that the overexpression of *PAB1*, *CDC5*, and *RGS2* mitigates the sensitivity of *gpn2*[ts] strains to temperature fluctuations, DNA damage agents, and transcription inhibitors (Fig 5A). However, this suppression is limited to semi-restrictive temperatures, as these genes cannot prevent death at lethal temperatures. Additionally, experiments conducted on *gpn1*[ts] strains, which are sensitive to both high and low temperatures, revealed that overexpression of *CDC5* reduced their sensitivity to low

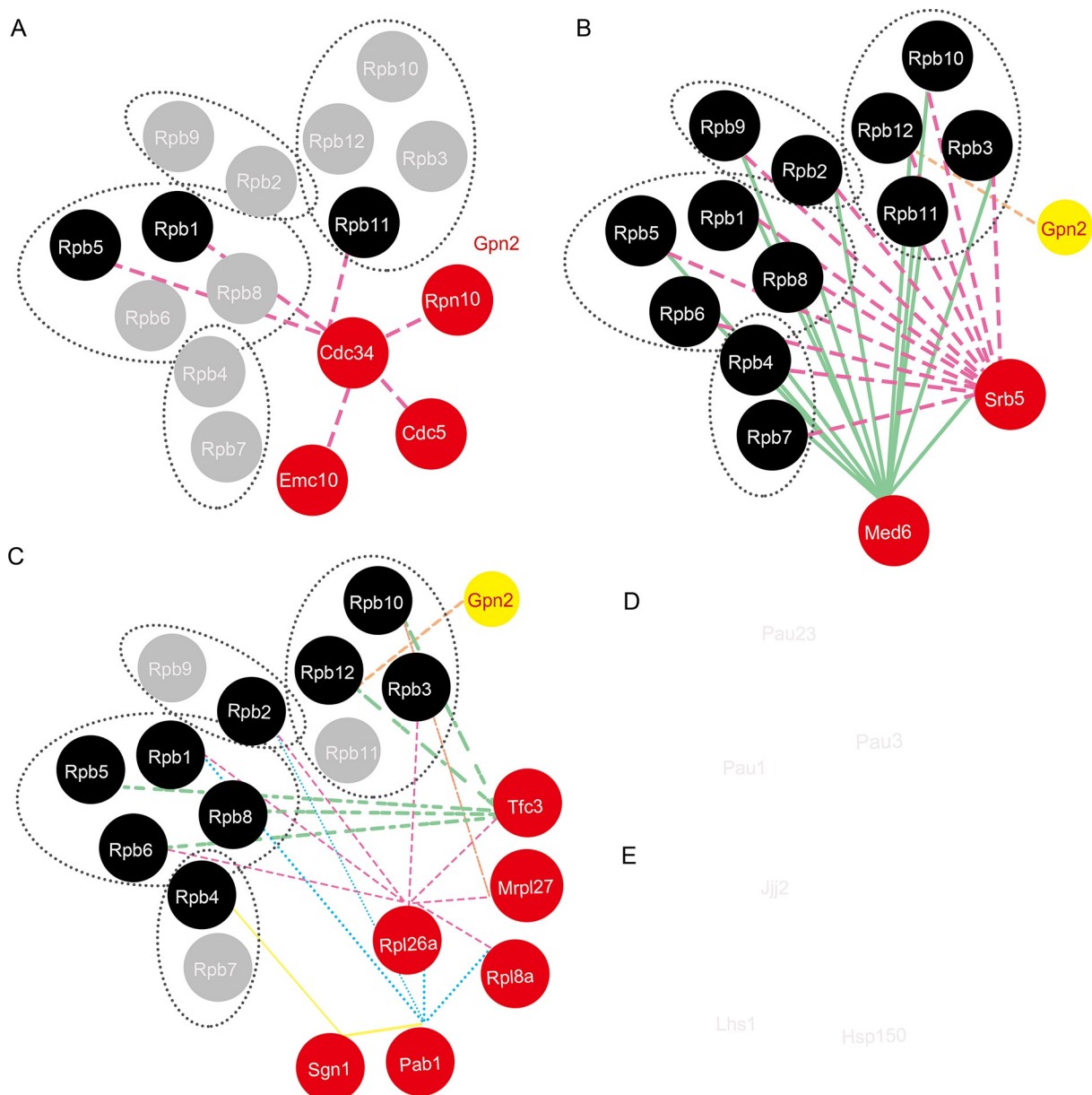

**Fig 4. Interactions between the suppressors and RNAPII subunits.** A-E: Network representation of interactions between suppressors and RNAPII subunits or among suppressors, visualized using Cytoscape. Known physical interactions are grouped by function, with connecting edge thickness and intensity indicating interaction strength (data from STRING, https://string-db.org). Node colors: red, suppressors; yellow, *GPN2*; all other nodes (black and gray in panels A-E) represent RNAPII subunits.

temperatures, DNA damage drugs, and transcriptional repression drugs. Similarly, the overexpression of *PAB1*, *CDC5*, and *RGS2* decreased the sensitivity of *gpn2*$^{ts}$ strains to semi-limiting temperatures (Fig 5B). High expression of these genes substantially compensated for the growth defects of *gpn2*$^{ts}$ in the presence of DNA-damaging and transcription-inhibitory drugs (Fig 5B). Given that the Gpn family proteins were initially characterized in chromosomal instability (CIN) mutants [28], it is plausible that *PAB1*, *CDC5*, and *RGS2* play crucial roles in regulating Gpn1 and Gpn2 activities that are essential for maintaining genome stability and transcriptional regulation.

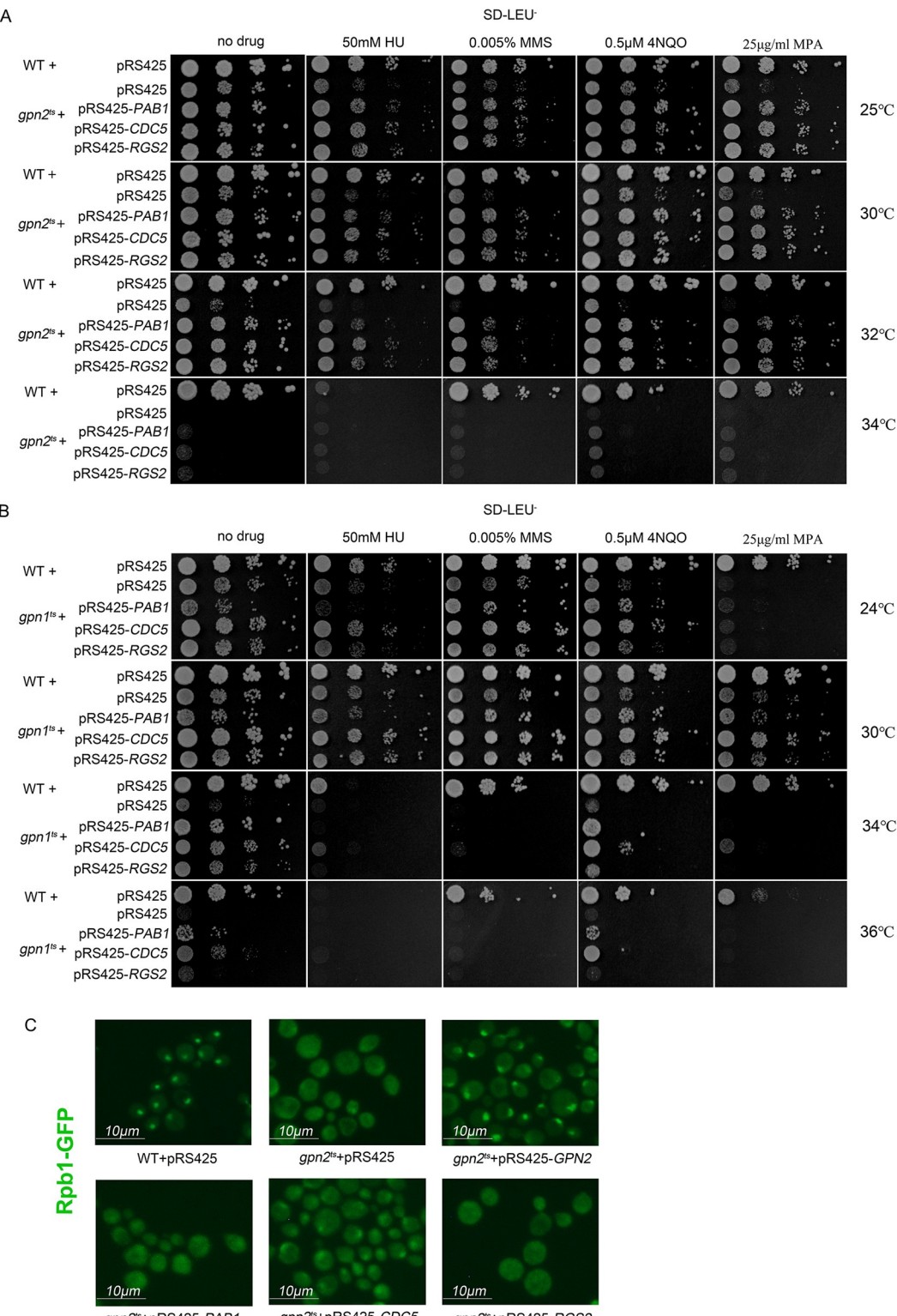

**Fig 5. *PAB1*, *CDC5*, and *RGS2* can alleviate the growth defects of *gpn1^ts* and *gpn2^ts* in the presence of DNA-damaging and transcriptional suppressors drugs.** A: Overexpression of *PAB1*, *CDC5*, and *RGS2* reduces the sensitivity of *gpn2^ts* strains to DNA-damaging agents. Tenfold serial dilutions of logarithmic-phase cultures were plated on SD-Leu⁻ plates containing hydroxyurea (HU), methyl methane sulfonate (MMS), 4-nitroquinoline oxide (4-NQO), or mycophenolic acid (MPA) at the indicated concentrations and temperatures, and incubated for four days. B: Similarly, overexpression of *PAB1*,

*CDC5*, and *RGS2* reduces the sensitivity of *gpn1^{ts}* strains to DNA-damaging agents. C: Overexpression of *CDC5* reduces the fluorescence diffusion of Rpb1 in *gpn2^{ts}* strains. Logarithmic-phase cultures were grown in SD-Leu⁻ medium at 34˚C for three hours, fixed with ethanol, and observed for fluorescence.

Previous studies have demonstrated that Gpn2 is involved in RNAPII assembly [9]. To delve deeper into the functions of *PAB1*, *CDC5*, and *RGS2* in relation to this process, we examined their impact on *gpn2^{ts}* mutants. Our findings revealed that elevated expression of *CDC5* mitigated the fluorescence dispersion of Rpb1, the largest subunit of RNAPII, indicating an effective stabilization or localization effect, whereas *RGS2* and *PAB1* showed no such influence (Fig 5C). Although overexpression of *CDC5* does not completely restore the nuclear localization of Rpb1, it underscores the close association between Cdc5 and Gpn2, suggesting a potential role of *CDC5* in modulating RNAPII assembly dynamics influenced by Gpn2.

## Discussion

Through our high-throughput screening, we identified a range of genes whose overexpression can suppress the defects observed in *gpn2^{ts}* strains. The comprehensive nature of the screen allowed us to identify a substantial number of candidate genes, each of which was subsequently confirmed through meticulous re-cloning and re-testing processes. Notably, several genomic regions, such as *CDC34*, were isolated multiple times independently, bolstering our confidence in the reliability and significance of our findings concerning genes that interact genetically with *GPN2*.

Further analysis of the suppressor interaction network revealed five distinct modules, illustrating diverse genetic interactions. Some suppressors were found to interact with subunits of RNAPII (Fig 4). Specifically, one module (Fig 4A) contains regulatory proteins; Fig 4B and 4C focus on transcription and translation-related factors. Additionally, other modules include members of the seripauperin multigene family [78] (Fig 4D) and chaperone proteins (Fig 4E). These interactions were delineated from manually curated clusters within the physical protein-protein interaction (PPI) database, providing valuable insights into the functions of these genes.

Our findings enable us to infer the functions of previously uncharacterized genes such as *EMC10* and *JJJ2*. *EMC10* is shown to interact with *CDC34* in the PPI database (Fig 4A). Given that *CDC34* also acts as a dosage suppressor of *gpn2^{ts}*, we propose that *EMC10* may function as a regulator of Cdc34. Overexpression of *EMC10* may thus modulate Cdc34 activity to compensate for the defective *GPN2*. *JJJ2*, featuring a J-domain homologous to the *E. coli* DnaJ molecular chaperone [79], interacts with the chaperone protein Lhs1 (Fig 4E). This suggests that Jjj2, along with Hsp150, Nas2, and Lhs1, may function as novel RNAPII assembly factors, similar to the roles of Hsp90/Hsp82 and the R2TP/prefoldin-like complex in the assembly of RNA polymerases [28]. In addition, we identified Rgs2, a GTPase-activating protein (GAP) that may regulate Gpn2 through its GAP activity.

Previous studies have demonstrated genetic and physiological interactions among the three members of the Gpn family, where *GPN1* and *GPN3* can suppress *gpn2^{ts}* mutants [9]. In this experiment, *CDC5* suppresses both *gpn2^{ts}* and *gpn1^{ts}* mutants (Fig 5A and 5B). Furthermore, it complements the fluorescence diffusion of Rpb1 in the nucleus (Fig 5C), suggesting that *CDC5* may also play an important role in transcriptional regulation. *CDC5* is an essential gene, and its overexpression causes phenotypes such as cell cycle arrest, actin cytoskeleton abnormalities, and other defects. As a protein kinase, it plays a critical regulatory role [66, 67]. Whether a reduction in Cdc5 enzyme activity can compensate for GPN mutants remains an open question, and its connection with the GPN family warrants further investigation.

There appear to be numerous ways in which the temperature-sensitive (TS) phenotype can be bypassed. Our results suggest that RNA polymerase activity, ribosome biogenesis, and DNA repair are linked to the functional defect in *gpn2^ts* mutants. Additionally, enhanced transcription, translation, and nuclear export can bypass the TS phenotype, possibly by influencing effector proteins. We believe that the findings from this multicopy suppressor screening will offer valuable insights into the functions of Gpn2 and its associated biological pathways.

## Supporting information

**S1 Raw data.**
(ZIP)

## Acknowledgments

We thank all members of the laboratory for their contributions. We also appreciate the assistance in sequencing from our biotechnology collaborators.

## Author Contributions

**Conceptualization:** Le Wang, Pan Li, Pei Zeng, Debao Xie, Mengdi Gao, Aamir Sohail, Fanli Zeng.

**Data curation:** Le Wang, Pei Zeng, Debao Xie.

**Formal analysis:** Pan Li, Pei Zeng, Aamir Sohail.

**Funding acquisition:** Fanli Zeng.

**Methodology:** Le Wang, Pei Zeng, Fanli Zeng.

**Project administration:** Le Wang, Pan Li, Fanli Zeng.

**Validation:** Mengdi Gao, Lujie Ma, Aamir Sohail.

**Visualization:** Le Wang, Pei Zeng.

**Writing – original draft:** Le Wang, Pan Li, Pei Zeng.

**Writing – review & editing:** Pan Li, Mengdi Gao, Lujie Ma, Aamir Sohail.

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
