## [Decision Letter · Decision Letter 0]

11 Jul 2024

PONE-D-24-22767Dosage suppressors of gpn2ts mutants and multiple functional speculations of Gpn2 in budding yeastPLOS ONE

Dear Dr. Wang,

Thank you for submitting your manuscript to PLOS ONE. After careful consideration, we feel that it has merit but does not fully meet PLOS ONE’s publication criteria as it currently stands. Therefore, we invite you to submit a revised version of the manuscript that addresses the points raised during the review process. I do believe releasing the screening data would largely benefit the scientific community. Nevertheless, both reviewers raised several concerns that require further attention.

We look forward to receiving your revised manuscript.

Kind regards,

Jorge Perez-Fernandez, Ph.D.

Academic Editor

PLOS ONE

https://academic.oup.com/g3journal/article/3/5/917/6025691?login=false

In your revision ensure you cite all your sources (including your own works), and quote or rephrase any duplicated text outside the methods section. Further consideration is dependent on these concerns being addressed.

4. Please include your tables as part of your main manuscript and remove the individual files. Please note that supplementary tables (should remain/ be uploaded) as separate "supporting information" files.

Reviewers' comments:

Reviewer's Responses to Questions

**Comments to the Author**

1. Is the manuscript technically sound, and do the data support the conclusions?

Reviewer #1: Yes

Reviewer #2: Yes

2. Has the statistical analysis been performed appropriately and rigorously? 

Reviewer #1: N/A

Reviewer #2: I Don't Know

3. Have the authors made all data underlying the findings in their manuscript fully available?

Reviewer #1: Yes

Reviewer #2: Yes

4. Is the manuscript presented in an intelligible fashion and written in standard English?

Reviewer #1: No

Reviewer #2: Yes

5. Review Comments to the Author

Reviewer #1: The GPN family members are a fascinating yet poorly understood family of proteins and a suppression screen is a valid approach to enhance our knowledge about Gpn2 function. The technical aspects of this submission are fine, using standard methods for genetic suppression screening and validation.

Comments:

Figure 1 A - this domain should read GTP binding, not ATP binding as written.

Figure 3 contains several spot assays to confirm that genetic suppression of the gpn2 mutant is occurring. However, some genes described in the results section such as MRPL27 and RFC1 have little to no effect on growth. These should be removed from the list of positive suppressors OR growth curve data should be generated to confirm that there is improved growth due to expression of these marginal suppressors.

Table 1 is a good summary of the results from this screen and more concise than the genes addressed in the text.

This read like a list making exercise and many of the descriptions of protein function listed in the results section are taken verbatim from the Yeast Genome Database pages for these genes which borders on plagarism.

I would have liked to see some attempt to further investigate how these suppressors affect growth of the gpn2 mutant cells. An easy experiment would be to look at RNAPII subunit localization with a GFP tagged protein as these are readily available in the yeast community.

The RGS2 suppressor result is interesting and should have been followed up upon. Perhaps transforming the RGS2 plasmid into gpn1 or gpn3 mutants to assess whether it's GAP activity restores growth of these mutants at elevated temperatures.

There are considerable issues with phrasing, economical use of words and grammar and the authors would benefit from having someone with strong written English skills review their writing before submission.

Overall, this feels like the start of an interesting project on Gpn2 function but without any followup experiments to elucidate how suppression of the growth phenotype occurs, it feels unfinished.

Reviewer #2: In this manuscript, the authors conduct a genetic suppression screen to identify possible suppressors of the gpn2ts mutation. Subsequently, they subclone the found suppressors to identify the genes responsible for correcting the growth of the mutant at high temperatures. Finally, the authors propose a series of possible functions for Gpn2 based on the results found. The experimental design and execution are adequate for the study's purposes. However, there are several aspects that should be taken into account.

First, the authors previously published a similar genetic suppression screen on the gpn2ts mutant, where they described the suppressors Rpb12, Rba50, and other GPNs (Zeng et al, 2018). Therefore, they should clarify whether this study is a continuation of that one. If so, they should name the RPB12 and RBA50 genes as suppressors, indicating that their function has been previously studied. If the study presented in this manuscript is a different screen, the authors should clarify why they do not find RPB12 and RBA50 as suppressors in this case.

Similarly, the sensitivity of the gpn2ts mutant to mycophenolic acid had been previously published and should therefore be indicated in the text.

The authors should clarify why they analyse the sensitivity of gpn2ts mutant to the other drugs but then do not use them to check the suppressors. They only look the suppression of three plasmids in DNA-damaging agents. They should look the suppression in the presence of all the studied drugs, as the effect of different drugs might vary depending on the suppressor gene, which could help clarify the function that the gene is correcting.

As minor corrections, in the third paragraph of the introduction, the authors write “interaction between Rpb12, Rpb3, and Rba50 subunits.” However, Rba50 is not a subunit of RNA pol II but an associated factor, so the wording should be corrected.

The beginning of the third paragraph of the introduction is confusing and should be rewritten: “In the preceding work conducted, it was found that Gpn2, one of the vital factors in RNA polymerase II assembly, is directly participating in the interaction between Rpb12, Rpb3, and Rba50 subunits. On the contrary, genetic screening data also confirmed the association of Gpn2 with Rba50 and Rpb3, which are the second-largest subunits of RNA polymerase II assembly factors [9].”

The work is understandable and experimentally well-structured. While it is in a very preliminary phase, it could be a considerable contribution if certain aspects mentioned above are clarified.

6. PLOS authors have the option to publish the peer review history of their article (what does this mean?). If published, this will include your full peer review and any attached files.

Reviewer #1: No

Reviewer #2: No

---

## [Author Response · Author response to Decision Letter 0]

24 Sep 2024

Dear Editor,

Thank you for providing us with the opportunity to resubmit our manuscript. We are delighted to know that it has been received positively by the reviewers and is provisionally accepted pending successful revision. Please find attached our revised manuscript along with the updated figures. We have made every effort to fully address the critiques provided, which are outlined in our detailed responses to the reviewers.

Reviewer #1: The Gpn family members are a fascinating yet poorly understood family of proteins and a suppression screen is a valid approach to enhance our knowledge about Gpn2 function. The technical aspects of this submission are fine, using standard methods for genetic suppression screening and validation.

Comments-1: Figure 1A - this domain should read GTP binding, not ATP binding as written.

Response 1: Thank you for your suggestion. We have revised “ATP” to “GTP” in Figure 1A.

Comments-2: Figure 3 contains several spot assays to confirm that genetic suppression of the gpn2 mutant is occurring. However, some genes described in the results section such as MRPL27 and RFC1 have little to no effect on growth. These should be removed from the list of positive suppressors or growth curve data should be generated to confirm that there is improved growth due to expression of these marginal suppressors.

Response 2: Thank you for your suggestion. Some genes described in the results section, such as MRPL27 and RFC1, exhibit minimal or no effect on growth at 34°C. However, they can compensate growth defects of the gpn2ts mutants at 32°C. Therefore, we consider these genes as candidate suppressors, as detailed in Table 1.

Comments-3: This read like a list making exercise and many of the descriptions of protein function listed in the results section are taken verbatim from the Yeast Genome Database pages for these genes which borders on plagarism.

Response 3: Thank you for pointing this out. We have significantly revised the manuscript to include additional research and our original insights. Please refer to L156-L176, L178-190, L195-L215, L217-L227, L229-L245, L247-L273.

Comments-4: I would have liked to see some attempt to further investigate how these suppressors affect growth of the gpn2 mutant cells. An easy experiment would be to look at RNAPII subunit localization with a GFP tagged protein as these are readily available in the yeast community.

Response 4: Your suggestions have been invaluable. We tagged Rpb1, the largest subunit of RNA polymerase II, with GFP in the gpn2ts mutant and subsequently overexpressed it in high-copy suppressors PAB1, CDC5, and RGS2 at 34°C. We found that CDC5 significantly compensated for the nuclear dispersion observed in the mutant (Figure 5C, L329-338).

Comments-5: The RGS2 suppressor result is interesting and should have been followed up upon. Perhaps transforming the RGS2 plasmid into gpn1 or gpn3 mutants to assess whether it's GAP activity restores growth of these mutants at elevated temperatures.

Response 5: Thank you for your valuable suggestion. We have transformed the high-expression constructs of PAB1, CDC5, and RGS2 into the gpn1ts and gpn2ts mutants and cultured them at various temperatures. Our findings indicate that they can alleviate the growth defects of gpn2ts in the presence of DNA-damaging and transcriptional suppressors drugs at 32°C. Additionally, at 34°C, PAB1, CDC5, and RGS2 each significantly reduced the sensitivity of gpn1ts mutants on 4-NQO tablets, with CDC5 showing a particularly notable effect (Figure 5A and B, L314-323)

Comments-6: There are considerable issues with phrasing, economical use of words and grammar and the authors would benefit from having someone with strong written English skills review their writing before submission.

Response 6: Thank you for your suggestions. We have thoroughly addressed the issues raised and have made comprehensive revisions to the language of our manuscript. To ensure accuracy in word choice, grammar, and sentence structure, the revised manuscript has been meticulously edited by an experienced science editor from A&L Scientific Editing (www.alpublish.com), whose first language is English and who specializes in editing scientific papers authored by non-native English speakers. We sincerely hope that the revised manuscript meets your expectations. We are grateful for your consideration of our work and would like to express our heartfelt appreciation once again.

Reviewer #2:

Comments-1: First, the authors previously published a similar genetic suppression screen on the gpn2ts mutant, where they described the suppressors Rpb12, Rba50, and other GPNs (Zeng et al, 2018). Therefore, they should clarify whether this study is a continuation of that one. If so, they should name the RPB12 and RBA50 genes as suppressors, indicating that their function has been previously studied. If the study presented in this manuscript is a different screen, the authors should clarify why they do not find RPB12 and RBA50 as suppressors in this case.

Response 1: Thank you for your suggestion. This study builds upon previous research. Our research group has already described the suppressors Rba50, Rpb12, and other GPNs in a prior study (Zeng et al, 2018), as noted in the manuscript (L66-69). In addition, this study aims to identify novel inhibitors, thereby providing a more robust theoretical foundation for investigating the role of Gpn2 in RNA polymerase II activity. 

Response 2: Similarly, the sensitivity of the gpn2ts mutant to mycophenolic acid had been previously published and should therefore be indicated in the text.

Response 2: Thank you for your suggestion. We have documented the sensitivity of the gpn2ts mutant to mycophenolic acid in the revised manuscript (L44-45).

Response 3: The authors should clarify why they analyse the sensitivity of gpn2ts mutant to the other drugs but then do not use them to check the suppressors. They only look the suppression of three plasmids in DNA-damaging agents. They should look the suppression in the presence of all the studied drugs, as the effect of different drugs might vary depending on the suppressor gene, which could help clarify the function that the gene is correcting.

Response 3: Thank you for your valuable suggestions. In response, we have added experiments to assess the sensitivity of the transcriptional suppressor mycophenolic acid (MPA) to gpn1ts and gpn2ts strains. This further elucidates that the suppressors PAB1, CDC5, and RGS2 are effective not only against DNA-damaging drugs but also at the transcriptional level. CDC5 exhibited the most pronounced inhibitory effect. Please refer to Figures 5A and B. Details are provided in the manuscript (L314-325).

Comments-4: As minor corrections, in the third paragraph of the introduction, the authors write “interaction between Rpb12, Rpb3, and Rba50 subunits.” However, Rba50 is not a subunit of RNA pol II but an associated factor, so the wording should be corrected.

Response 4: Thank you for your suggestion. We have corrected this in our manuscript (L41-42).

Comments-5: The beginning of the third paragraph of the introduction is confusing and should be rewritten: “In the preceding work conducted, it was found that Gpn2, one of the vital factors in RNA polymerase II assembly, is directly participating in the interaction between Rpb12, Rpb3, and Rba50 subunits. On the contrary, genetic screening data also confirmed the association of Gpn2 with Rba50 and Rpb3, which are the second-largest subunits of RNA polymerase II assembly factors.”

Response 5: Thank you for pointing out the confusion. We have revised the beginning of the third paragraph of the introduction for clarity (L40-43).

---

## [Decision Letter · Decision Letter 1]

19 Oct 2024

PONE-D-24-22767R1Dosage Suppressors of gpn2 ts  Mutants and Functional Insights into the Role of Gpn2 in Budding Yeast

PLOS ONE

Dear Dr. Wang,

Thank you for submitting your manuscript to PLOS ONE. After careful consideration, we feel that it has merit but does not fully meet PLOS ONE’s publication criteria as it currently stands. Therefore, we invite you to submit a revised version of the manuscript that addresses the points raised during the review process.

Please, address the few concerns raised by reviewers and submit your revised manuscript by Dec 03 2024 11:59PM. If you will need more time than this to complete your revisions, please reply to this message or contact the journal office at plosone@plos.org. Please include the following items when submitting your revised manuscript:

We look forward to receiving your revised manuscript.

Kind regards,

Jorge Perez-Fernandez, Ph.D.

Academic Editor

PLOS ONE

Journal Requirements:

Reviewers' comments:

Reviewer's Responses to Questions

**Comments to the Author**

1. If the authors have adequately addressed your comments raised in a previous round of review and you feel that this manuscript is now acceptable for publication, you may indicate that here to bypass the “Comments to the Author” section, enter your conflict of interest statement in the “Confidential to Editor” section, and submit your "Accept" recommendation.

Reviewer #1: All comments have been addressed

Reviewer #2: All comments have been addressed

2. Is the manuscript technically sound, and do the data support the conclusions?

Reviewer #1: Yes

Reviewer #2: Yes

3. Has the statistical analysis been performed appropriately and rigorously? 

Reviewer #1: N/A

Reviewer #2: Yes

4. Have the authors made all data underlying the findings in their manuscript fully available?

Reviewer #1: Yes

Reviewer #2: Yes

5. Is the manuscript presented in an intelligible fashion and written in standard English?

Reviewer #1: Yes

Reviewer #2: Yes

6. Review Comments to the Author

Reviewer #1: PLOS One notes re-submission PONE-D-24-22767R1

Text edits were noted and appreciated as it strengthens the manuscript.

It is nice to see the use of "old school" suppression screens to uncover new biology.

I do hope that the authors or another group follows up on this work as the GPN proteins deserved to be more well understood given their expression across species.

Figure 2 and 4 not included in the resubmission. I am assuming there were no changes from the original manuscript.

Minor text changes:

Figure 5. I would remove the wording "significantly" from the statement about CDC5 overexpression restoring Rpb1 localization in the gpn2 mutant as there was no quantitative analysis done. However, the qualitative result is fairly obvious.

Methods section should include a line about Rpb1-GFP whether it was plasmid based or integrated into the genome.

Reviewer #2: (No Response)

7. PLOS authors have the option to publish the peer review history of their article (what does this mean?). If published, this will include your full peer review and any attached files.

Reviewer #1: No

Reviewer #2: No

---

## [Author Response · Author response to Decision Letter 1]

24 Oct 2024

Dear Editor,

We are very pleased to know that our manuscript was received positively by the reviewers and that it is provisionally accepted, pending successful minor revision. Please find attached our revised manuscript. We have made every effort to fully address the critiques, which are outlined in our replies to reviewers. Please note the itemized responses below, where the original comments are underlined in black and our responses are displayed in blue. We sincerely hope that the updated version meets your expectations. We are grateful for your consideration of our publication and would like to express our heartfelt appreciation once more.

Reviewer #1: 

Figure 2 and 4 not included in the resubmission. I am assuming there were no changes from the original manuscript.

Figure 5. I would remove the wording "significantly" from the statement about CDC5 overexpression restoring Rpb1 localization in the gpn2 mutant as there was no quantitative analysis done. However, the qualitative result is fairly obvious.

Methods section should include a line about Rpb1-GFP whether it was plasmid based or integrated into the genome.

Comments-1: Thank you very much for your valuable feedback. Figures 2 and 4 were no changes from the original manuscript. We have thoroughly reviewed and revised the entire manuscript, removing the wording "significantly" from the statement about CDC5 overexpression restoring Rpb1 localization in the gpn2 mutant. Additionally, we have included a description of Rpb1-GFP in the Materials and Methods section (L88-91).

---

## [Editor Report · Decision Letter 2]

29 Oct 2024

Dosage Suppressors of gpn2 ts  Mutants and Functional Insights into the Role of Gpn2 in Budding Yeast

PONE-D-24-22767R2

Dear Dr. Wang,

We’re pleased to inform you that your manuscript has been judged scientifically suitable for publication and will be formally accepted for publication once it meets all outstanding technical requirements.

Kind regards,

Jorge Perez-Fernandez, Ph.D.

Academic Editor

PLOS ONE

---

## [Editor Report · Acceptance letter]

26 Nov 2024

PONE-D-24-22767R2 

PLOS ONE

Dear Dr. Wang, 

I'm pleased to inform you that your manuscript has been deemed suitable for publication in PLOS ONE. Congratulations! Your manuscript is now being handed over to our production team.

Kind regards, 

on behalf of

Dr. Jorge Perez-Fernandez 

Academic Editor

PLOS ONE